# Separation of benzene and toluene associated with vapochromic behaviors by hybrid[4]arene-based co-crystals

Jingyu Chen[1], Wenjie Zhang[1], Wenzhi Yang[1], Fengcheng Xi[1], Hongyi He[1], Minghao Liang[1], Qian Dong[1], Jiawang Hou[1], Mengbin Wang [2] ✉, Guocan Yu [3] ✉ & Jiong Zhou [1] ✉

The combination of macrocyclic chemistry with co-crystal engineering has promoted the development of materials with vapochromic behaviors in supramolecular science. Herein, we develop a macrocycle co-crystal based on hybrid[4]arene and 1,2,4,5-tetracyanobenzene that is able to construct vapochromic materials. After the capture of benzene and toluene vapors, activated hybrid[4]arene-based co-crystal forms new structures, accompanied by color changes from brown to yellow. However, when hybrid[4]arene-based co-crystal captures cyclohexane and pyridine, neither structures nor colors change. Interestingly, hybrid[4]arene-based co-crystal can separate benzene from a benzene/cyclohexane equal-volume mixture and allow toluene to be removed from a toluene/ pyridine equal-volume mixture with purities reaching 100%. In addition, the process of adsorptive separation can be visually monitored. The selectivity of benzene from a benzene/cyclohexane equal-volume mixture and toluene from a toluene/ pyridine equal-volume mixture is attributed to the different changes in the charge-transfer interaction between hybrid[4]arene and 1,2,4,5-tetracyanobenzene when hybrid[4]arene-based co-crystal captures different vapors. Moreover, hybrid[4]arene-based co-crystal can be reused without losing selectivity and performance. This work constructs a vapochromic material for hydrocarbon separation.

Vapochromism is a behavior that causes the color of a compound or material to change due to trapping vapors[1]. Materials with vapochromic behavior can reflect the specific environment well through the change of color, and have been widely used in environmental monitoring, optoelectronic devices, gas sensing and so on[2–5]. In recent years, co-crystal has provided a unique strategy for the preparation of multifunctional vapochromic materials[6]. Co-crystal is a crystalline monophasic material containing two or more components, which is co-crystallized by non-covalent interactions, such as charge-transfer interaction, π···π interaction, hydrogen bond and C–H···X interactions

(X: halogen)[7,8]. At present, co-crystal materials play important roles in fields of photoelectric applications and pharmaceutical engineering[9–11]. In recent years, macrocycle co-crystals have been widely investigated, which have broadened the range of vapochromic materials[12–15]. Macrocycle-based vapochromic materials realize the response of vapors through the vapor-triggered solid-state structural transformation to form charge-transfer co-crystals[16–18]. In addition, macrocycle co-crystals are also suitable for constructing solid-state microlasers and thermally activated delayed fluorescence materials[19,20].

[1]Department of Chemistry, College of Sciences, Northeastern University, Shenyang 110819, PR China. [2]ZJU-Hangzhou Global Scientific and Technological Innovation Center, Zhejiang University, Hangzhou 311215, PR China. [3]Key Laboratory of Bioorganic Phosphorus Chemistry & Chemical Biology, Department of Chemistry, Tsinghua University, Beijing 100084, PR China. ✉e-mail: 21737055@zju.edu.cn; guocanyu@mail.tsinghua.edu.cn; zhoujiong@mail.neu.edu.cn

Precise synthesis of substances often requires high-purity solvents to reduce the proportion of by-products[21]. Benzene (Bz) and its homologs are commonly used solvents and play important roles in synthetic chemistry and chemical industry[22]. Cyclohexane (Cy) is directly formed by the hydrogenation of Bz[23]. Unreacted Bz reduces the purity of Cy[24]. Subtle differences in structures and boiling points (Bz: 80.2 °C; Cy: 80.7 °C) prevent them from being separated by traditional distillation. Thus, the separation and purification of Bz and Cy mixtures have been extensively studied and are considered as one of the most challenging tasks in the chemical industry[25]. In addition, toluene (Tol) is one of the important homologs of Bz. Tol shares many of the same properties with Bz and is often utilized as an organic solvent in place of the latter, which is of considerable toxicity. In industry, Tol is obtained mainly from coal and petroleum through catalytic reforming, hydrocarbon pyrolysis and coking[26,27]. However, the Tol product obtained from the coal coking process often contains a small quantity of pyridine (Py)[28]. In order to obtain high-purity Tol, the small amount of Py present in the Tol during coal coking process must be removed[29]. However, the similar boiling points of Tol and Py (Tol: 110.6 °C; Py: 115.2 °C) make purification challenging[30]. To date, a variety of techniques have been employed to separate mixtures of Bz and Cy or Tol and Py, such as extractive distillation and pressure-swing distillation[31]. However, these techniques usually need large operating costs. Therefore, it is urgent to develop an energy-saving and efficient separation method.

Taking use of the variations in molecular geometry and size, adsorptive separation via porous materials is an effective technique[32,33]. Porous materials such as metal−organic frameworks (MOFs), covalent organic frameworks (COFs), porous organic polymers (POPs) and zeolites have been extensively studied for adsorption and separation due to their low energy consumption and high Brunauer−Emmett−Teller (BET) specific surface areas[34–37]. For example, one of the prominent aspects of MOFs is the ability to modulate the size and shape of their pores[38]. However, their industrial applications are limited by factors such as low stability and complex design[39]. Thus, it is still urgent to develop a kind of absorbent with stable structure and high selectivity.

During the past few decades, the research of macrocyclic hosts has developed rapidly. The abundant molecular recognition and self-assembly characteristics of macrocycles make them widely used in drug delivery, sensor, adsorptive separation and so on[40–45]. In recent years, macrocycles-based nonporous adaptive crystals (NACs) materials have been flourishing[46,47]. NACs are nonporous in their original crystalline states, however, particular vaporized species can induce to produce intrinsic pores through supramolecular interactions, resulting in cavities to catch guest molecules[48–50]. In addition, compared with traditional porous materials, NACs have the advantages of good thermal stability, easy preparation and moisture resistance. Hybridarenes are relatively new class of macrocycles, which are synthesized by two or more different types of units[51]. Hybridarenes broaden the synthesis of macrocyclic hosts with varying shapes, cavity sizes, and flexibilities. Recently, hybridarenes-based NACs, as a class of nonporous and structurally adaptable crystalline materials, have attracted widespread attention for their advantages in adsorption and separation, such as the highly selective separation of Bz and Cy by hybrid[3]arene-based NACs[52]. However, the vapochromic properties of hybridarenes have yet to be developed. To our knowledge, there are no reports of hybridarenes-based co-crystals being used as adsorptive materials for industrial separation.

Herein, we report the synthesis and structure of hybrid[4]arene (H). A hybrid[4]arene-based co-crystal (H-TCNB) is prepared by charge-transfer interaction between electron-rich macrocycle H and electron-deficient 1,2,4,5-tetracyanobenzene (TCNB) (Fig. 1). Activated co-crystal (H-TCNB$\alpha$), as adaptive macrocycle co-crystal adsorptive separation materials, permit the removal of Bz from Bz and Cy

mixtures ($v$:$v$, 1:1) as well as Tol from Tol and Py mixtures ($v$:$v$, 1:1) with purities reaching 100%. Meanwhile, H-TCNB$\alpha$ display vapochromic behaviors for Bz and Tol, accompanied by color changes from brown to yellow and crystalline phase structural transformation. Furthermore, the adsorbed Bz and Tol can be removed by heating, so that these co-crystals return to their original states, endowing H-TCNB$\alpha$ with distinguished recovery performance.

## Results

### Synthesis of hybrid[4]arene
The macrocyclic host hybrid[4]arene H was successfully synthesized through a simple one-step reaction. To prepare H, a mixture of 1 equiv. of 2,6-diethoxylnaphthalene (DON), 1 equiv. of 1,3-dimethoxybenzene (DOB) and 2 equiv. of paraformaldehyde with a catalytic amount of trifluoroacetic acid (TFA) were refluxed in CHCl$_3$ (Fig. 1a). The reaction progress was monitored by thin-layer chromatography. After the reaction was finished, a saturated aqueous solution of Na$_2$CO$_3$ was added to neutralize TFA. The product was purified by column chromatography to give H as a white solid with a yield of 50%. The structure of H was determined by $^1$H NMR, $^{13}$C NMR, ESI-MS (Supplementary Figs. 1–4) and single crystal X-ray diffraction (Fig. 2).

### Single crystal structure of H
Single crystals of H suitable for X-ray analysis were grown through slow diffusion of $n$-hexane into CH$_2$Cl$_2$ solution of H at room temperature. H was made of two 2,6-diethoxylnaphthalene units with two 1,3-dimethoxybenzene units linked by methylene bridges (Fig. 2a). H exhibited a neat quadrilateral-prismatic structure. Each unit sat on a different plane, with the two naphthalene units parallel to each other and the two benzene units nearly orthogonal (Fig. 2b). The crystal stacking model of H showed that the adjacent H was compacted in the opposite direction into a cross-layer structure similar to a brick wall (Supplementary Fig. 5). In addition, each H was arranged parallel to its neighboring H through intermolecular C−H···π and C−H···O interactions (Supplementary Figs. 6 and 7).

### Theoretical calculation of H
Moreover, the chemical structure and electrostatic potential (ESP) map of H showed that the naphthalene walls of H were highly electronegative (Fig. 2c). The result demonstrated the possibility of the external host–guest complexation. In order to better understand the electronic structures of H, density functional theory (DFT) calculations were conducted. The frontier molecular orbital diagrams demonstrated that the highest occupied molecular orbital (HOMO) of H was concentrated on the electron-donating naphthalene units, with a small distribution on the benzene units (Fig. 2d). Besides, the lowest unoccupied molecular orbital (LUMO) of H was totally localized on the naphthalene units. These observations suggested that the formation of co-crystal would benefit from the naphthalene units.

### Characterization of Hα
The activated crystal H (H$\alpha$) was obtained by vacuum drying at 150 °C for 10 h. Thermogravimetric analysis (TGA) proved that the solvent in H$\alpha$ was removed (Supplementary Fig. 8). Powder X-ray diffraction (PXRD) suggested that H$\alpha$ was crystalline (Supplementary Fig. 9). N$_2$ adsorption experiment showed that H$\alpha$ was nonporous, with a BET specific surface area of 0.820 m$^2$/g (Supplementary Fig. 10).

### Preparation and characterization of co-crystal
The matching donor (H) and the chosen acceptor (TCNB) formed co-crystal (denoted as H-TCNB) in CH$_2$Cl$_2$ driven through charge-transfer interaction (Fig. 3a). The complexation of H with TCNB in solution was investigated by NMR and UV-$vis$ absorption spectroscopy. After adding TCNB to H in CD$_2$Cl$_2$, the signal related to the proton H$_a$ on TCNB shifted upfield ($\Delta\delta = -0.12$ ppm). The signals related to protons H$_1$, H$_4$, H$_5$, H$_6$,

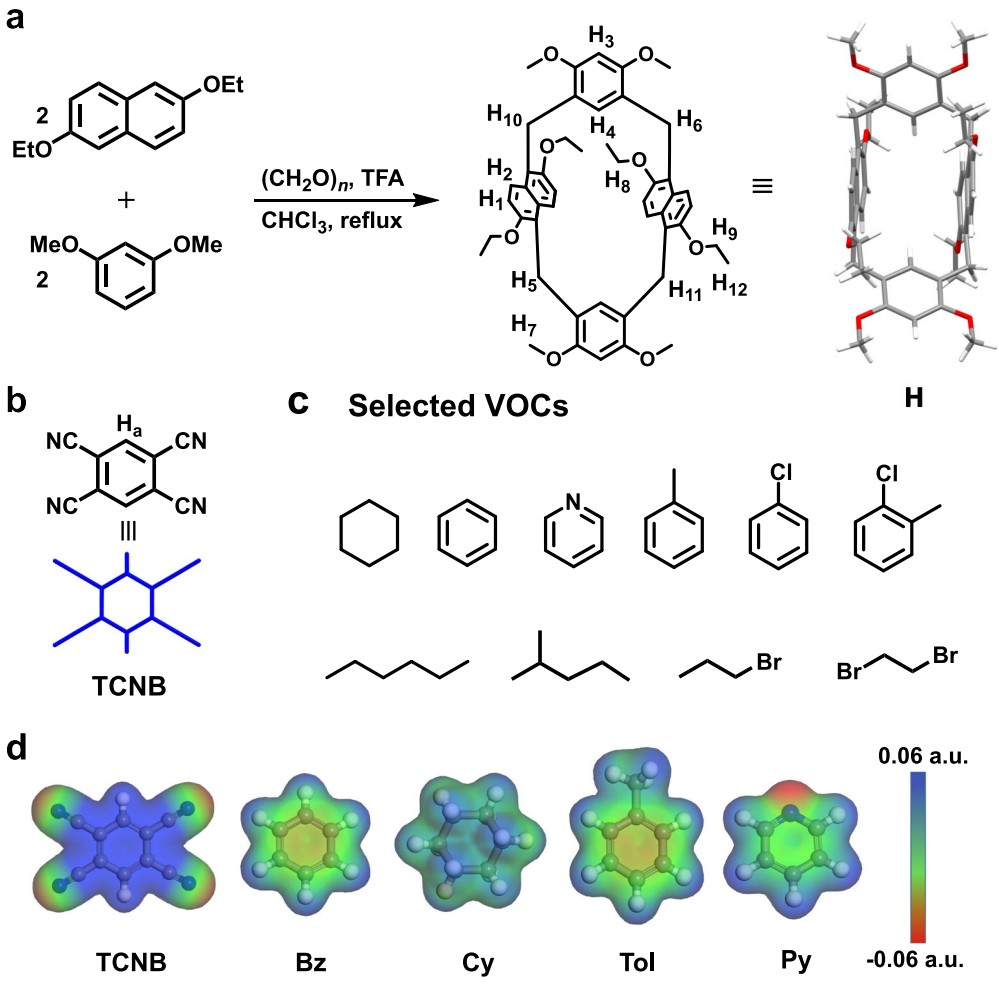

**Fig. 1 | Chemical structures and electrostatic potential maps. a** Synthesis of hybrid[4]arene (H). Chemical structures of: **b** 1,2,4,5-tetracyanobenzene (TCNB); **c** selected volatile organic compounds (VOCs). **d** Electrostatic potential (ESP) maps of TCNB, Bz, Cy, Tol, and Py.

$H_{10}$ and $H_{11}$ on H shifted upfield after complexation ($\Delta\delta = -0.06$, $-0.02$, $-0.03$, $-0.02$, $-0.03$, $-0.03$ ppm, respectively), indicating a fast exchange process (Supplementary Figs. 11 and 12)[18]. Besides, in the 2D NOSEY spectrum of H-TCNB, the NOE signals between protons $H_1$, $H_2$, $H_3$ and $H_{12}$ on H and proton $H_a$ on TCNB proved the complexation (Supplementary Figs. 13 and 14). After adding TCNB to H in $CH_2Cl_2$, the color of the solution of H changed from colorless to purple, and a new absorption band appeared at 400 nm in the UV-*vis* absorption spectrum. This phenomenon confirmed the existence of a strong charge-transfer interaction between H and TCNB (Figs. 4a, b). Furthermore, when TCNB was added to H in $CDCl_3$, the signal related to the proton $H_a$ on TCNB shifted upfield slightly ($\Delta\delta = -0.04$ ppm). The NOE signals between protons $H_2$, $H_{10}$ and $H_{11}$ on H and proton $H_a$ on TCNB in the 2D NOSEY spectrum demonstrated the weak non-covalent interactions (Supplementary Figs. 15-18). Therefore, the formation of co-crystal in $CH_2Cl_2$ was more suitable for the construction of vapochromic materials.

To further confirm the interaction between H and TCNB, a single crystal (denoted as H-TCNB@$CH_2Cl_2$) was obtained by slowly diffusing *n*-hexane into a $CH_2Cl_2$ solution of H and TCNB. As shown in Fig. 4f, H, TCNB and $CH_2Cl_2$ were combined in a ratio of 1:1:1 to form H-TCNB@$CH_2Cl_2$ (Supplementary Fig. 19). Each TCNB molecule was parallel to the naphthalene ring of H molecule and was connected to another adjacent H molecule by multiple C–H···π and C–H···N interactions (C–H···π distances: 3.359 Å, 3.365 Å and 3.385 Å; C–H···N distances: 2.610 Å and 2.616 Å), forming an ordered co-crystal structure (Supplementary Fig. 20). In addition, H molecule was interleaved along the crystallographic *a*-axis. Each TCNB molecule was sandwiched

between four H molecules (Supplementary Fig. 21). In the stacking structure of H-TCNB@$CH_2Cl_2$, the $CH_2Cl_2$ and TCNB molecules filled the gaps of empty 1D channels along the crystallographic *c*-axis formed by H (Supplementary Fig. 22). Interestingly, the $CH_2Cl_2$ molecule acted as the bridge between TCNB molecule and H molecule. The C–H···O interaction between hydrogen atoms of $CH_2Cl_2$ molecule and oxygen atoms of H molecule (C–H···O distances: 2.419 Å and 2.343 Å) stabilized H and $CH_2Cl_2$ (Supplementary Fig. 23). In addition, there existed C–H···Cl interaction between hydrogen atoms of H molecule and Cl atoms of $CH_2Cl_2$ molecule (C–H···Cl distance: 2.936 Å). Each TCNB and $CH_2Cl_2$ molecules were stabilized by C–H···N interaction (C–H···N distances: 2.342 Å and 3.218 Å). Furthermore, each TCNB molecule was embraced by four $CH_2Cl_2$ molecules, forming a parallelogram-like structure (Supplementary Fig. 24).

### Characterization of H-TCNBα

To obtain adsorptive and vaporchromic materials, H-TCNB@$CH_2Cl_2$ was activated by heating at 90 °C for 10 h under vacuum (named H-TCNBα). [1]H NMR and TGA verified that $CH_2Cl_2$ was completely removed from H-TCNBα, and the molar ratio of H to TCNB was 1:1 (Supplementary Figs. 26 and 27). The BET specific surface area of H-TCNBα was determined to be 0.970 m²/g by $N_2$ sorption experiment, revealing that H-TCNBα was nonporous (Supplementary Fig. 28). The differential scanning calorimetric (DSC) study showed the melting and decomposition temperatures of H-TCNBα, which was revealed by two endothermic peaks at 230 °C and 260 °C, respectively (Supplementary Fig. 29). The PXRD pattern of H-TCNBα was similar to the experimental

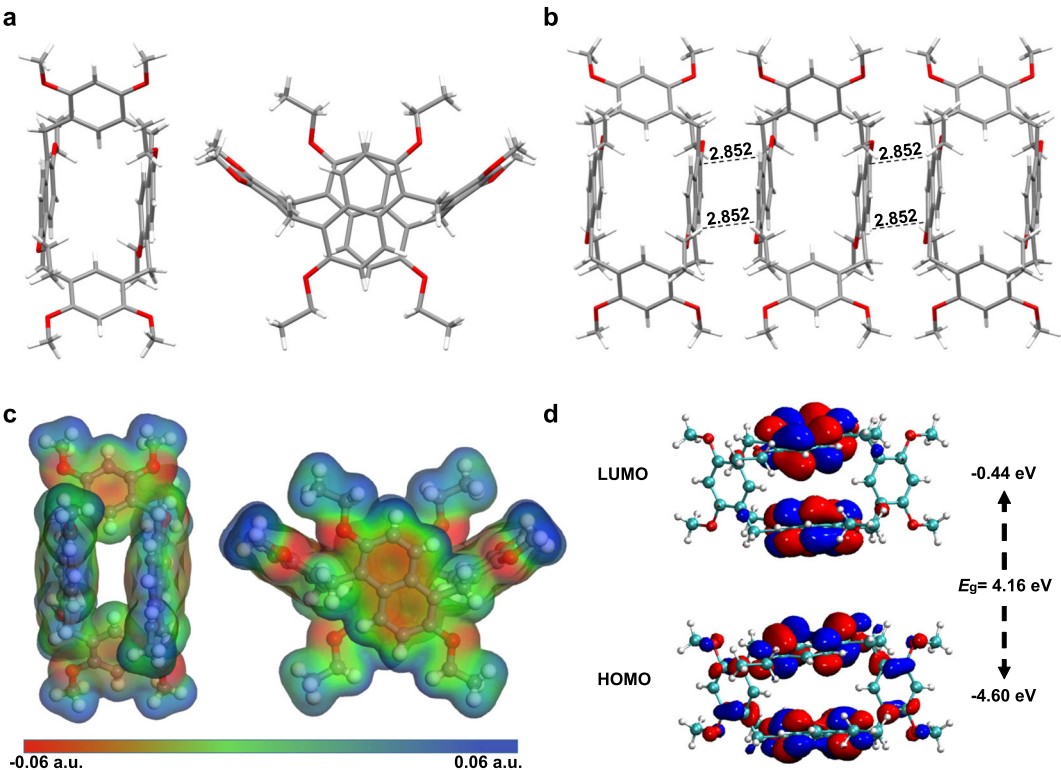

**Fig. 2 | Characterization of H. a** Single crystal structure of H (Top view and side view). **b** Molecular stacking model of H in the crystalline state. **c** ESP maps of H (top view and side view). **d** HOMO and LUMO orbital distributions of H.

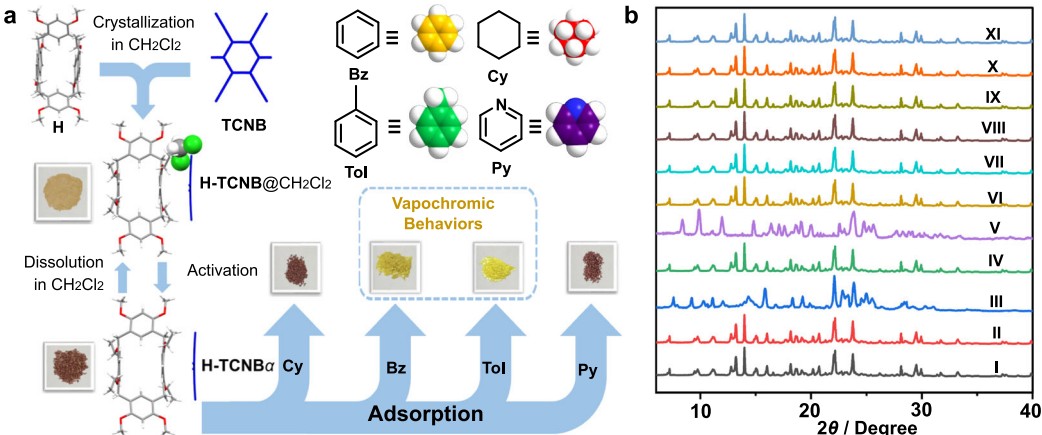

**Fig. 3 | A general strategy for constructing vapochromic system toward VOCs. a** Structures of hybrid[4]arene (H), 1,2,4,5-tetracyanobenzene (TCNB), benzene (Bz), cyclohexane (Cy), toluene (Tol) and pyridine (Py), and schematic representation of vapochromic behaviors of H-TCNBα after exposure to Cy, Bz, Tol and Py vapors. **b** PXRD patterns of H-TCNBα before (I) and after capture of (II) Cy, (III) Bz, (IV) Py, (V) Tol, (VI) chlorobenzene, (VII) 2-chlorotoluene, (VIII) *n*-hexane, (IX) 2-methylpentane, (X) BrCH₂CH₃ and (XI) BrCH₂CH₂Br.

and simulated PXRD patterns of H-TCNB@CH₂Cl₂ (Fig. 4c). The ESP map of H-TCNB@CH₂Cl₂ was similar to that of H-TCNBα, further confirming that the structure and charge distribution of co-crystal was maintained during the activation process (Figs. 4g, h). Optical and scanning electron microscopic (SEM) studies suggested that H and TCNB formed sheet structures. In addition, H-TCNBα formed bulk lamellar structure (Fig. 4e). The Fourier transform infrared spectrum (FT-IR) of TCNB showed a typical stretching vibration peak associated with the CN group of TCNB at 2247 cm⁻¹ (Fig. 4d). However, the CN peak in H-TCNBα was redshifted, demonstrating the charge-transfer interaction between TCNB and H. The diffuse reflectance spectrum of H-TCNBα exhibited an obvious broad band at 550 nm (Supplementary

Fig. 30), while no absorption was observed for both H and TCNB. In addition, the Raman spectrum of H-TCNBα showed that the representative characteristic peak (around 2250 cm⁻¹) related to the CN group of TCNB completely disappeared (Supplementary Fig. 31). These results demonstrated that the multiple charge-transfer interactions between H and TCNB in H-TCNBα might result in a shielding effect.

### Adsorptive and vapochromic behaviors of single-component Bz and Cy or Tol and Py vapors by H-TCNBα

The color of H-TCNBα changed significantly from brown to yellow when exposed to Bz or Tol vapor for 24 h (Figs. 5a and d, insert). In addition, the uptake of Bz vapor was calculated as one Bz vapor per

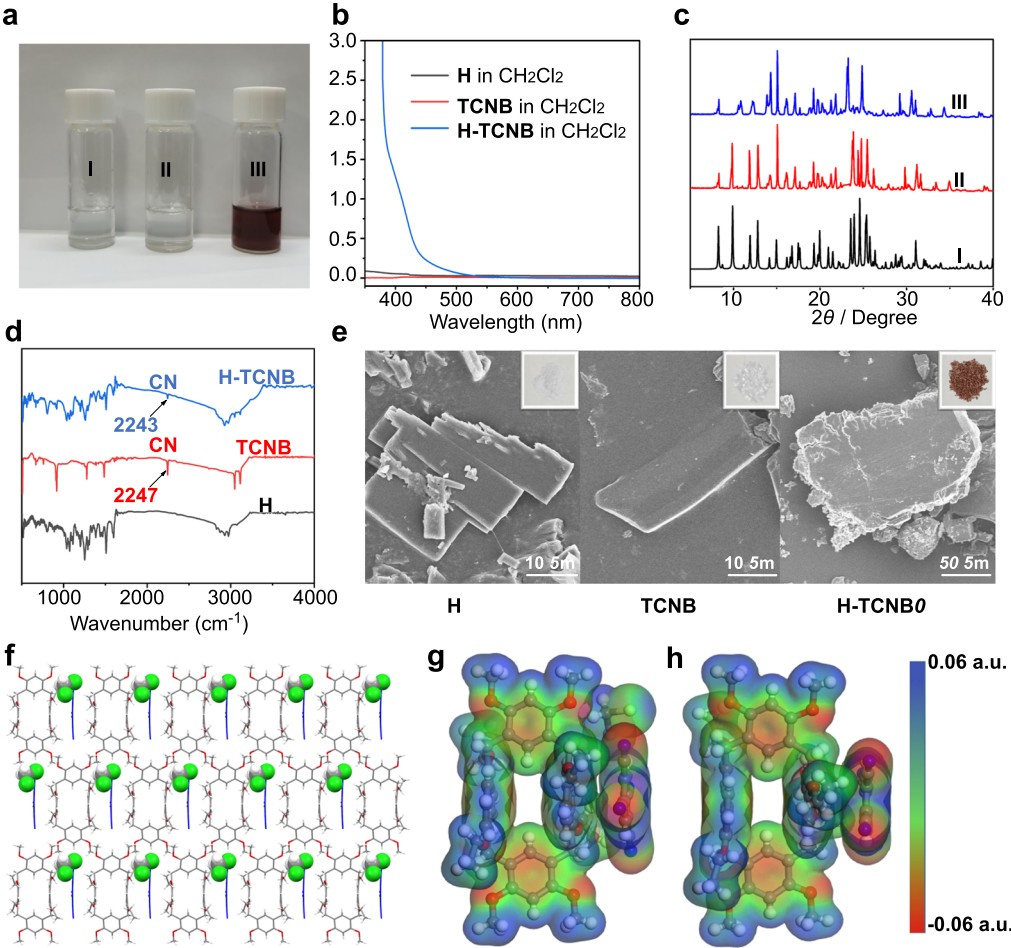

**Fig. 4 | Characterization of co-crystal. a** Images: (I) H (5.00 mM) in $CH_2Cl_2$; (II) TCNB (5.00 mM) in $CH_2Cl_2$; (III) H (5.00 mM) and TCNB (5.00 mM) in $CH_2Cl_2$. **b** UV-*vis* spectra: H (5.00 mM); TCNB (5.00 mM); H (5.00 mM) and TCNB (5.00 mM). **c** PXRD patterns: (I) simulated from the single crystal structure of H-TCNB@$CH_2Cl_2$; (II) H-TCNB@$CH_2Cl_2$; (III) H-TCNB$α$. **d** FT-IR spectra of H, TCNB and H-TCNB. **e** SEM images of H, TCNB and H-TCNB$α$. Insets showed the corresponding solid powders. **f** The stacking model of H-TCNB@$CH_2Cl_2$ (Special color treatments for clarity. Color codes: C, light gray; H, white; O, red; Cl, green; TCNB, blue). ESP maps of **g** H-TCNB@$CH_2Cl_2$ and **h** H-TCNB$α$.

H-TCNB molecule at saturation by [1]H NMR spectrum (Fig. 5a, black line; Supplementary Fig. 32). The uptake of Tol vapor was calculated as 0.9 Tol vapor per H-TCNB molecule at saturation by [1]H NMR spectrum (Fig. 5d, black line; Supplementary Fig. 34). On the contrary, the uptake of Cy or Py vapor by H-TCNB$α$ was negligible (Supplementary Figs. 33 and 35). The PXRD patterns of H-TCNB$α$ significantly changed after the adsorption of Bz vapor, but did not change after the adsorption of Cy vapor (Fig. 5b), meaning that the capture of Bz vapor caused the emergence of a new crystalline structure. TGA of H-TCNB$α$ after adsorption of Bz vapor showed a weight loss of 8.3% at 100 °C, indicating that one H-TCNB$α$ molecule accommodated one Bz molecule (Supplementary Fig. 36). However, there was almost no weight loss before 200 °C of H-TCNB$α$ upon exposure to Cy vapor for 24 h (Supplementary Fig. 37). In addition, the PXRD patterns of H-TCNB$α$ changed after adsorption of Tol vapor, but did not change after uptake of Py vapor (Fig. 5e), meaning that the capture of Tol vapor caused the emergence of a new crystalline structure. TGA of H-TCNB$α$ after adsorption of Tol vapor showed a weight loss of 8.7% at 90 °C, indicating that one H-TCNB$α$ molecule accommodated 0.86 Tol molecule (Supplementary Fig. 38). But there was almost no weight loss before 200 °C after adsorption of Py vapor by H-TCNB$α$ for 24 h (Supplementary Fig. 39). These results showed that H-TCNB$α$ could capture Bz or Tol vapor, but not Cy or Py vapor.

Diffuse reflectance spectroscopy was used to study the vapo-chromic behavior of H-TCNB$α$. After exposure to Bz or Tol vapor, the diffuse reflectance spectra of H-TCNB$α$ showed an obvious blue shift over time (Fig. 5c, f), along with the color of H-TCNB$α$ changed from brown to yellow. Compared with H-TCNB$α$, the adsorption bands at 700 nm almost disappeared in H-TCNB@Bz and H-TCNB@Tol (Supplementary Figs. 40 and 41). By contrast, the adsorption band of H-TCNB$α$ exposed to Cy or Py vapor was consistent with that of H-TCNB$α$. The result showed that the charge-transfer interaction between H and TCNB could be changed by Bz or Tol vapor, making H-TCNB$α$ a vapochromic material with Bz and Tol vapors selectivity.

Furthermore, typical stretching vibration peaks associated with the CN group of TCNB were observed for both H-TCNB@Bz and H-TCNB@Tol at 2247 cm$^{-1}$ in FT-IR spectra. However, compared with the corresponding stretching vibration peak of H-TCNB$α$, the typical absorption peaks of H-TCNB@Cy and H-TCNB@Py at 2237 cm$^{-1}$ were redshifted (Supplementary Figs. 42 and 43). The result verified that the structure of H-TCNB$α$ was changed after exposure to Bz or Tol vapor. Moreover, the Raman spectrum of H-TCNB$α$ showed that the representative characteristic peak (around 2250 cm$^{-1}$) associated with the CN group of TCNB disappeared, possibly due to the shielding effect caused by the charge-transfer interaction between H and TCNB in H-TCNB$α$. Similarly, the representative characteristic peaks associated with the CN group of TCNB were absent in Raman spectra of H-TCNB@Cy and H-TCNB@Py. The result demonstrated that the charge-transfer interaction between H and TCNB in H-TCNB$α$ was maintained after exposure to Cy or Py vapor. On the contrary, the Raman spectra

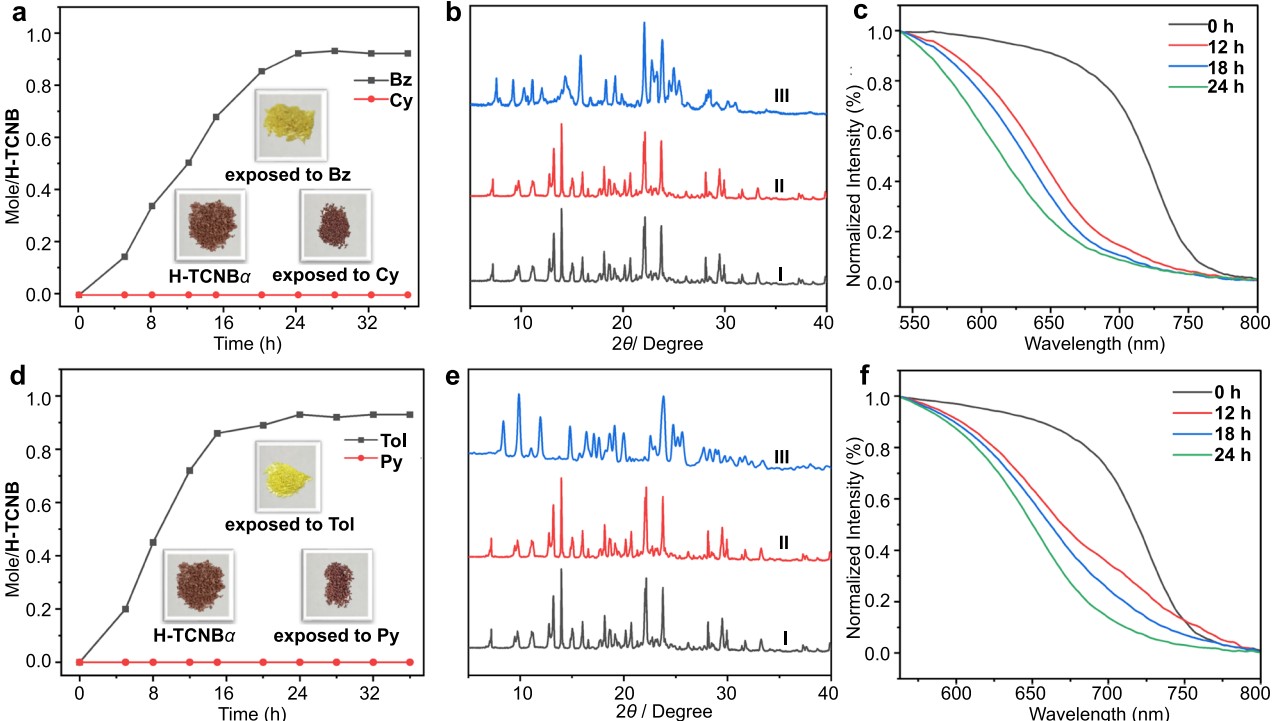

**Fig. 5 | Adsorption and vapochromic behaviors of H-TCNBα toward single-component VOCs. a** Time-dependent solid–vapor sorption plots of H-TCNBα for single-component Bz and Cy vapors. Inset showed the color changes of H-TCNBα after adsorption of Bz and Cy vapors. **b** PXRD patterns of H-TCNBα: (I) original H-TCNBα, (II) after adsorption of Cy vapor, (III) after adsorption of Bz vapor. **c** Normalized time-dependent diffuse reflectance spectra of H-TCNBα exposed to Bz vapor. **d** Time-dependent solid–vapor sorption plots of H-TCNBα for single-component Tol and Py vapors. Inset showed the color changes of H-TCNBα after adsorption of Tol and Py vapors. **e** PXRD patterns of H-TCNBα: (I) original H-TCNBα, (II) after adsorption of Py vapor, (III) after adsorption of Tol vapor. **f** Normalized time-dependent diffuse reflectance spectra of H-TCNBα exposed to Tol vapor.

of H-TCNB@Bz and H-TCNB@Tol showed representative characteristic peaks (around 2250 cm$^{-1}$) that were consistent with that of free TCNB. The results indicated that Bz and Tol vapors could weaken the charge-transfer interaction between H and TCNB, so that the shielding effect disappeared (Supplementary Figs. 44 and 45). Besides, H-TCNBα also exhibited selective vapochromic behaviors for other volatile organic compounds (VOCs). For example, when H-TCNBα was exposed to chlorobenzene, 2-chlorotoluene, BrCH$_2$CH$_2$Br, CH$_3$CH$_2$Br, n-hexane, and 2-methylpentane vapors, no color changes were observed (Supplementary Fig. 46). The PXRD patterns of H-TCNBα did not change after absorption of these VOCs (Fig. 3b).

## Crystallographic data analysis of co-crystals

To investigate the mechanism of vapochromic behaviors and adsorption properties of H-TCNBα for Bz and Tol vapors, single crystals of H-TCNB@Bz and H-TCNB@Tol were grown. Single crystal of H-TCNB@Bz suitable for X-ray analysis was obtained by putting H and TCNB powders in a vial where 0.5 mL of Bz was added. However, the suitable single crystal of H-TCNB@Tol has failed after various attempts.

In the single crystal of H-TCNB@CH$_2$Cl$_2$, TCNB and H molecules were stabilized by C−H···π and π···π interactions, with the TCNB molecule parallel to the naphthalene ring of the H molecule (Fig. 6a, b). In the 1:1 exo-wall charge-transfer complex of H with TCNB, it was found that the interplanar distance was 3.267 Å and 3.372 Å, respectively, revealing the strong charge-transfer interaction between H and TCNB molecules (Supplementary Fig. 25). However, in the crystal structure of H-TCNB@Bz, the interplanar distance between TCNB molecule and its neighboring H molecule was different from that of H-TCNB@CH$_2$Cl$_2$ (Fig. 6c, f). The interplanar distance of H with TCNB in H-TCNB@Bz was 3.494 Å and 3.587 Å, respectively (Supplementary Fig. 48). In the process of vapor capture, the color change of materials

with vapochromic properties is easily affected by the rearrangement of donor and acceptor[37]. The Bz and TCNB molecules filled the gaps of empty 1D channels along the crystallographic c-axis formed by H in the stacking structure (Fig. 5d, e and Supplementary Fig. 47). Each TCNB molecule was parallel to a naphthalene ring of H molecule and was connected to another adjacent H molecule by multiple C−H···π interactions (C−H···π distances: 2.662 Å and 2.733 Å).

Notably, solvents (CH$_2$Cl$_2$ or Bz) in co-crystal systems exhibited diverse stacking models. Each TCNB and CH$_2$Cl$_2$ molecules were stabilized by C−H···N interaction, but in the single crystal of H-TCNB@Bz, each TCNB and Bz molecules were stabilized by C−H···π interaction (C−H···N distances: 2.342 Å and 3.218 Å; C−H···π distance: 2.799 Å, Supplementary Fig. 49). Thanks to the C−H···π interaction between TCNB and Bz molecules, supramolecular tessellation with a rhombic tiling pattern could be realized (Supplementary Fig. 50). In addition, the encapsulated Bz molecule was stabilized by multiple C−H···π and C−H···O interactions with H (C−H···π distances: 2.234 Å, 2.685 Å, 2.800 Å and 2.804 Å; C−H···O distances: 2.669 Å, 2.685 Å and 2.707 Å) (Supplementary Fig. 51). Briefly, Bz molecule acted as the linker of TCNB molecule with H molecule through charge-transfer interaction, facilitating the formation of co-crystals with effective stability.

## Selectivity experiments of H-TCNBα for mixture vapors of Bz and Cy or Tol and Py

The selective separation of Bz and Cy mixture vapors was facilitated by different vapochromic response of VOCs. Time-dependent solid-vapor adsorption experiments were performed by exposing H-TCNBα to Bz and Cy mixture vapors (v:v = 1:1). As shown in Fig. 7a, the absorption of Bz vapor by H-TCNBα gradually increased and reached saturation at around 20 h, accompanying by the color of H-TCNBα changed from brown to yellow. In addition, the saturation uptake of Bz vapor by

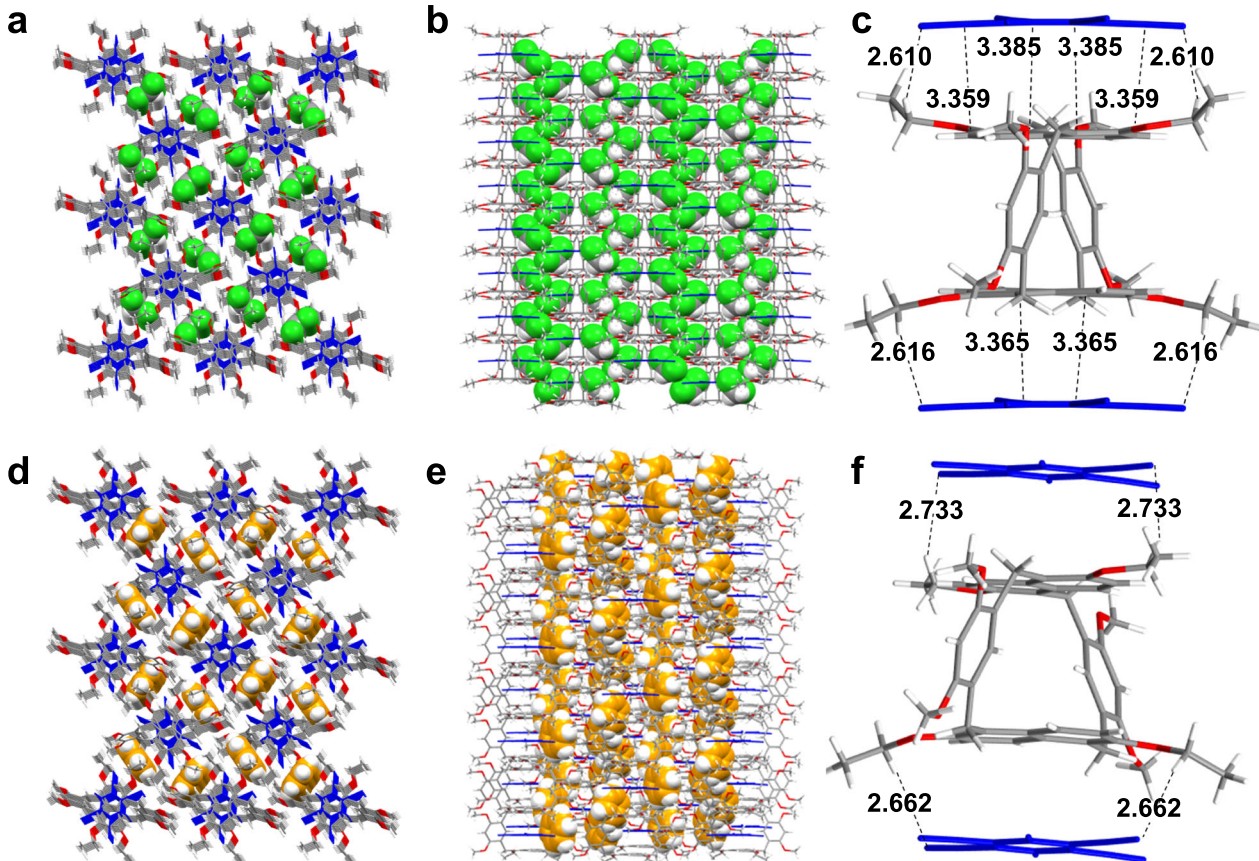

**Fig. 6 | Structural analysis of H-TCNB@CH₂Cl₂ and H-TCNB@Bz. a** Top view and **b** side view of stacking model of H-TCNB@CH₂Cl₂. **c** Charge-transfer interaction between H and TCNB in H-TCNB@CH₂Cl₂. **d** Top view and **e** side view of stacking model of H-TCNB@Bz. **f** Charge-transfer interaction between H and TCNB in H-TCNB@Bz. Included solvent molecules are omitted for clarity in (**c**) and (**f**).

H-TCNBα was nearly one Bz per H-TCNB. However, the amount of Cy vapor adsorbed by H-TCNBα was negligible (Supplementary Fig. 52). Gas chromatographic analysis showed that the content of Bz vapor was 100%, but there was no obvious adsorption for Cy vapor, confirming the high selectivity of H-TCNBα for Bz vapor (Supplementary Fig. 53). Moreover, the PXRD pattern of H-TCNBα upon uptake of Bz and Cy mixture vapors was consistent with that after adsorption of Bz vapor alone, and matched with that simulated from the single crystal data of H-TCNB@Bz (Fig. 7b). The results implied a structural transformation from H-TCNBα to H-TCNB@Bz upon the selective capture of Bz vapor from Bz and Cy mixture vapors.

In addition, FT-IR spectra of H-TCNBα after adsorption of Bz and Cy mixture vapors showed that the CN peak in H-TCNBα was red-shifted, similar to that of free TCNB (Fig. 7d and Supplementary Fig. 54). In the Raman spectra of H-TCNBα, a representative characteristic peak (around 2250 cm⁻¹) appeared when H-TCNBα was exposed to Bz and Cy mixture vapors (Supplementary Fig. 55). The results proved that Bz vapor in Bz and Cy mixture vapors played an important role in weakening the charge-transfer interaction between H and TCNB in H-TCNBα (Fig. 7e). This phenomenon was consistent with the result of the above-mentioned single-component experiment, further demonstrating the high selectivity of H-TCNBα for Bz vapor. ESP maps of H-TCNBα and H-TCNB@Bz showed that electrons in the original electron-deficient Bz were fully delocalized to the electron-rich area of methoxyl group on H upon adsorption (Fig. 7f). The results confirmed that the high selectivity was not only arose from the stability of the newly formed co-crystal structure after adsorbing guest, but also came from the strong charge-transfer interaction between H-TCNBα and Bz.

Furthermore, the competitive adsorption experiments of Bz and Cy vapors by H-TCNBα were performed to study the reversible structural transformation between H-TCNB@Cy and H-TCNB@Bz. When H-TCNB@Cy was exposed to Bz vapor, the color of H-TCNBα changed from brown to yellow (Fig. 7c). However, there was no obvious color change when H-TCNB@Bz was exposed to Cy vapor. The result suggested that the H-TCNB@Bz was more thermodynamically stable than the H-TCNB@Cy. Independent gradient model (IGM) analyses based on the optimized structures of H-TCNB@CH₂Cl₂ and H-TCNB@Bz clearly revealed the existence of multiple non-covalent interactions between H and TCNB (green regions, Fig. 8)[53]. As shown in Fig. 8a, the charge-transfer interaction between H and TCNB in H-TCNBα was stronger than that of between H and TCNB in H-TCNB@Bz. Besides, the non-covalent interaction analysis of H-TCNB@Bz showed a strong charge-transfer interaction between Bz and H (Fig. 8b). These results verified that H-TCNBα could selectively adsorb Bz from Bz and Cy mixtures, leading the structural transformation from H-TCNBα to H-TCNB@Bz.

The time-dependent selective separation experiment of H-TCNBα for Tol and Py mixture vapors was performed. H-TCNBα could gradually capture Tol vapor from Tol and Py mixture vapors (v:v = 1:1), and the color of H-TCNBα changed from brown to yellow (Fig. 9a). By contrast, the adsorption of Py vapor by H-TCNBα was negligible (Supplementary Fig. 56). These results indicated that H-TCNBα selectively adsorbed Tol, but not Py. Moreover, the PXRD pattern of H-TCNBα upon uptake of Tol and Py mixture vapors was consistent with that of H-TCNBα after adsorption of Tol vapor alone (Fig. 9b). Gas chromatographic analysis showed that the content of Tol was 100% (Supplementary Fig. 57). In addition, the CN peak of H-TCNBα in the

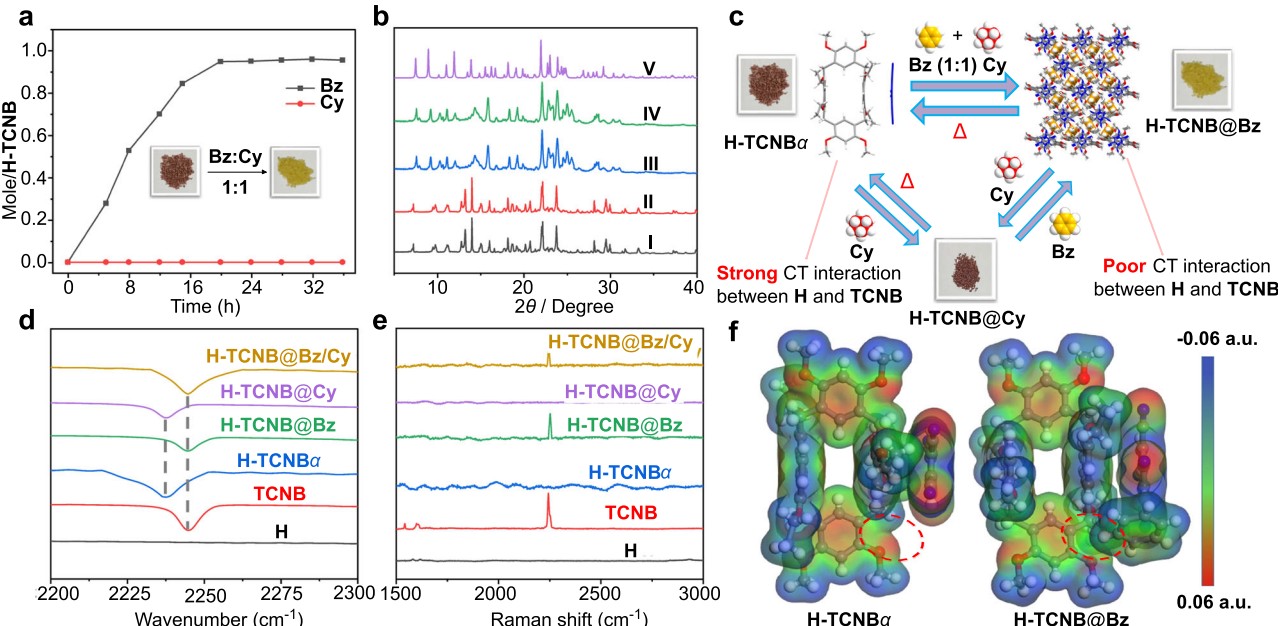

**Fig. 7 | Vapochromic behaviors H-TCNBα toward the vapors of Bz and Cy.**
**a** Time-dependent solid-vapor sorption plots of H-TCNBα for Bz and Cy mixture vapors (*v:v* = 1:1). Inset showed the color change of H-TCNBα after exposure to Bz and Cy mixture vapors. **b** PXRD patterns of H-TCNBα: (I) original H-TCNBα; (II) after adsorption of Cy vapor; (III) after adsorption of Bz vapor; (IV) after adsorption of Bz and Cy mixture vapors; (V) simulated from the single crystal structure of H-TCNB@Bz. **c** Schematic representation of the competitive adsorption experiment of H-TCNBα on Bz and Cy vapors. **d** Partial FT-IR spectra of H, TCNB, H-TCNBα, H-TCNB@Bz, H-TCNB@Cy and H-TCNB@Bz/Cy. **e** Partial Raman spectra of H, TCNB, H-TCNBα, H-TCNB@Bz, H-TCNB@Cy and H-TCNB@Bz/Cy. **f** ESP maps of H-TCNBα and H-TCNB@Bz. The area of red circles represents the change in electron density on H-TCNBα after the capture of Bz.

FT-IR spectrum was redshifted after adsorption of Tol and Py mixture vapors, similar to that of free TCNB (Fig. 9c and Supplementary Fig. 58). When H-TCNBα was exposed to Tol and Py mixture vapors, a characteristic peak (around 2250 cm⁻¹) associated with the CN group of TCNB appeared in the Raman spectrum (Fig. 9d and Supplementary Fig. 59). These results demonstrated that the charge-transfer interaction between H and TCNB could be weakened by Tol and Py mixture vapors, which was similar to using Tol alone. The above experiments showed that H-TCNBα could selectively adsorb Tol from Tol and Py mixture vapors.

In order to gain a deeper understanding on the selective adsorption of toluene/pyridine, DFT calculations were performed at the wB97XD/6−311 G(d) theoretical level, revealing the binding modes of H-TCNB to toluene (Tol) and pyridine (Py). In the calculated binding mode of H-TCNB@Py, the centroid-centroid distance of H with TCNB was 3.297 Å, which was similar to that of H-TCNBα (Supplementary Fig. 60). In the calculated binding mode of H-TCNB@Tol, the centroid-centroid distance of H with TCNB was 3.596 Å, which was different from that of H-TCNBα. The results showed that the structure of H-TCNBα changed after exposure to Tol vapor. The calculated energy values showed that the formation of H-TCNB@Tol was thermodynamically feasible ($\Delta H_{Sol}^{298}$ = −43.33 kJ/mol, $\Delta G_{Sol}^{298}$ = −38.33 kJ/mol) (Supplementary Table 4). Compared with H-TCNB@Tol, the thermodynamic energy values of H-TCNB@Py were higher ($\Delta H_{Sol}^{298}$ = 20.16 kJ/mol, $\Delta G_{Sol}^{298}$ = 12.20 kJ/mol), illustrating that the thermodynamic structure of H-TCNB@Tol was more stable. In agreement with the experimentally observed selectivity, the thermodynamic calculation results showed that the selectivity arose from the thermodynamic stability of the new formed co-crystals structure of H-TCNBα upon capturing the Tol.

## Purification experiments

Considering the highly selective adsorption of Bz and Tol vapors by H-TCNBα, we wondered whether H-TCNBα could be used to remove the trace amount of Bz from Bz and Cy mixture vapors or Tol from Tol and Py mixture vapors to obtain extremely pure Cy or

Py, respectively. For this purpose, H-TCNBα (3.00 mg) was exposed to 98.00% Cy (1.00 mL, containing 2.00% Bz impurity). After adsorption for 24 h, gas chromatographic analysis showed that the content of Bz reduced from 2.00% to 1.21%, and the content of Cy increased from 98.00% to 98.79% (Supplementary Figs. 61 and 62). In addition, when H-TCNBα (6.00 mg) was exposed to 98.00% Cy, gas chromatographic analysis showed that the content of Bz reduced from 2.00% to 0.57%, and the content of Cy increased from 98.00% to 99.43% (Supplementary Fig. 63). After exposing H-TCNBα (9.00 mg) to 98.00% Cy for 24 h, gas chromatographic analysis showed the content of Bz reduced from 2.00% to 0.28%. Importantly, the content of Cy increased significantly, reaching 99.72% (Fig. 10a and Supplementary Fig. 64). Hence, H-TCNBα could act as an adsorbent to remove the trace amount of Bz from Cy, resulting in extremely high purity and value Cy.

Furthermore, H-TCNBα was used to remove the trace amount of Tol from Tol and Py mixture vapors to produce extremely pure Py. H-TCNBα (3.00 mg) was exposed to 97.94% Py (1.00 mL, containing 2.06% Tol impurity). As verified by gas chromatography analysis, after adsorption for 24 h, the content of Tol reduced from 2.06% to 1.51%, and the purity of Py increased from 97.94% to 98.49% (Supplementary Figs. 65 and 66). After exposing H-TCNBα (6.00 mg) to 97.94% Py for 24 h, gas chromatographic analysis showed that the content of Tol reduced from 2.06% to 0.96% and the purity of Py increased from 97.94% to 99.04% (Supplementary Fig. 67). When H-TCNBα (9.00 mg) was exposed to 97.94% Py for 24 h, the content of Tol reduced from 2.06% to 0.65%, and the content of Py increased sharply to 99.35% (Fig. 10b and Supplementary Fig. 68). Therefore, H-TCNBα could be used as an adsorbent to obtain Py with extremely high purity and value.

## Recyclability experiments

The recyclability of the adsorbent is an essential factor in the actual industrial process[54]. H-TCNBα was regenerated by removing Bz and Tol at 120 °C under vacuum for 9 h. The newly obtained co-crystal was

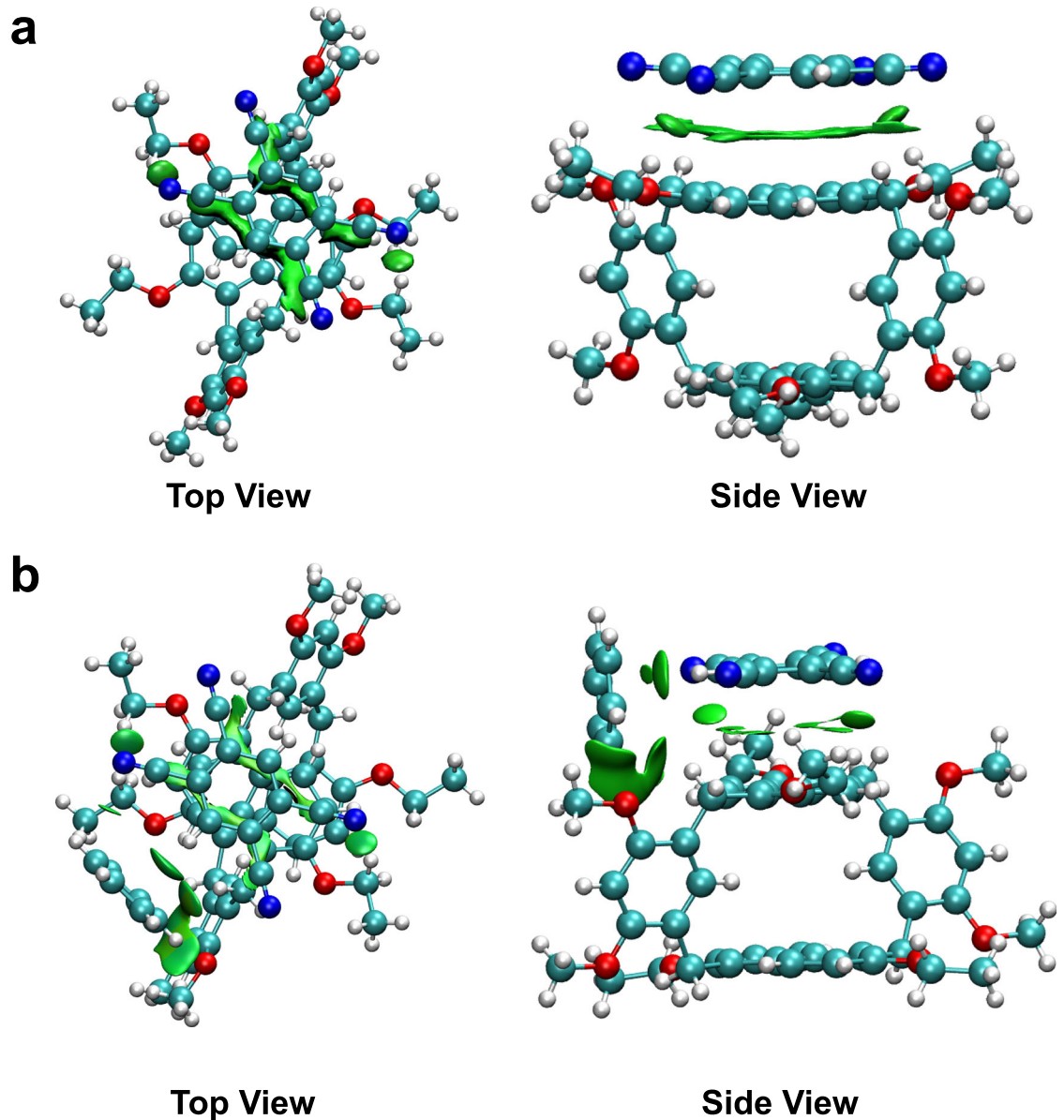

**Fig. 8 | Analysis of multiple non-covalent interactions.** The non-covalent interaction of **a** H-TCNBα and **b** H-TCNB@Bz. Green regions referred to | multiple non-covalent interactions between H and TCNB.

identified as H-TCNBα by ¹H NMR, PXRD, TGA and diffuse reflectance spectra, and H-TCNBα recovered its original brown color (Supplementary Figs. 69-74). Furthermore, H-TCNBα maintained high selectivity for Bz and Tol without losing performance for at least 5 cycles (Figs. 10c, d).

**Adsorptive and vapochromic behaviors of single-component Bz, Cy, Tol and Py vapors by DON-based co-crystals and DOB-based co-crystals**

In order to verify the adsorptive and vaporchromic behaviors of co-crystal in this study could only be achieved with a macrocyclic structure, the adsorptive and vapochromic behaviors by DON-based co-crystals and DOB-based co-crystals have been studied. To obtain adsorptive and vaporchromic materials, DON-TCNB was activated by heating at 80 °C for 10 h under vacuum (named DON-TCNBα) (Supplementary Figs. 75-84). The uptake of Bz, Cy, Tol or Py vapor by DON-TCNBα was negligible by ¹H NMR spectra (Supplementary Figs. 85-89). Besides, the PXRD patterns of DON-TCNBα did not change after adsorption of Bz, Cy, Tol or Py vapor (Supplementary Fig. 90). These results showed that DON-TCNBα could not capture Bz, Cy, Tol or Py vapor.

Using the same method, attempts were made to obtain the co-crystals of DOB-TCNB (Supplementary Figs. 91–93). When washed with ethanol, the color of the crystals changed from green to colorless (Supplementary Fig. 94). The colorless crystals were activated by heating at 80 °C for 10 h under vacuum (named DOB-TCNBα). ¹H NMR, PXRD, FT-IR verified that the colorless crystals were TCNB (Supplementary Figs. 95–97). Above all, the adsorptive and vaporchromic behaviors of hybrid[4]arene can only be achieved with a macrocyclic structure.

## Discussion

In summary, we have synthesized hybrid[4]arene (H), and developed a macrocycle co-crystal by exo-wall complexation of electron-rich macrocycle of H with electron-deficient TCNB. As macrocycle-based vapochromic materials, the color of H-TCNBα changed from brown to yellow as a result of the selective adsorption of Bz and Tol vapors. However, when H-TCNBα captured Cy and Py, neither structures nor colors changed. H-TCNBα could effectively separate Bz from Bz and Cy mixture vapors as well as Tol from Tol and Py mixture vapors with purities of 100%. In addition, the process of adsorptive separation

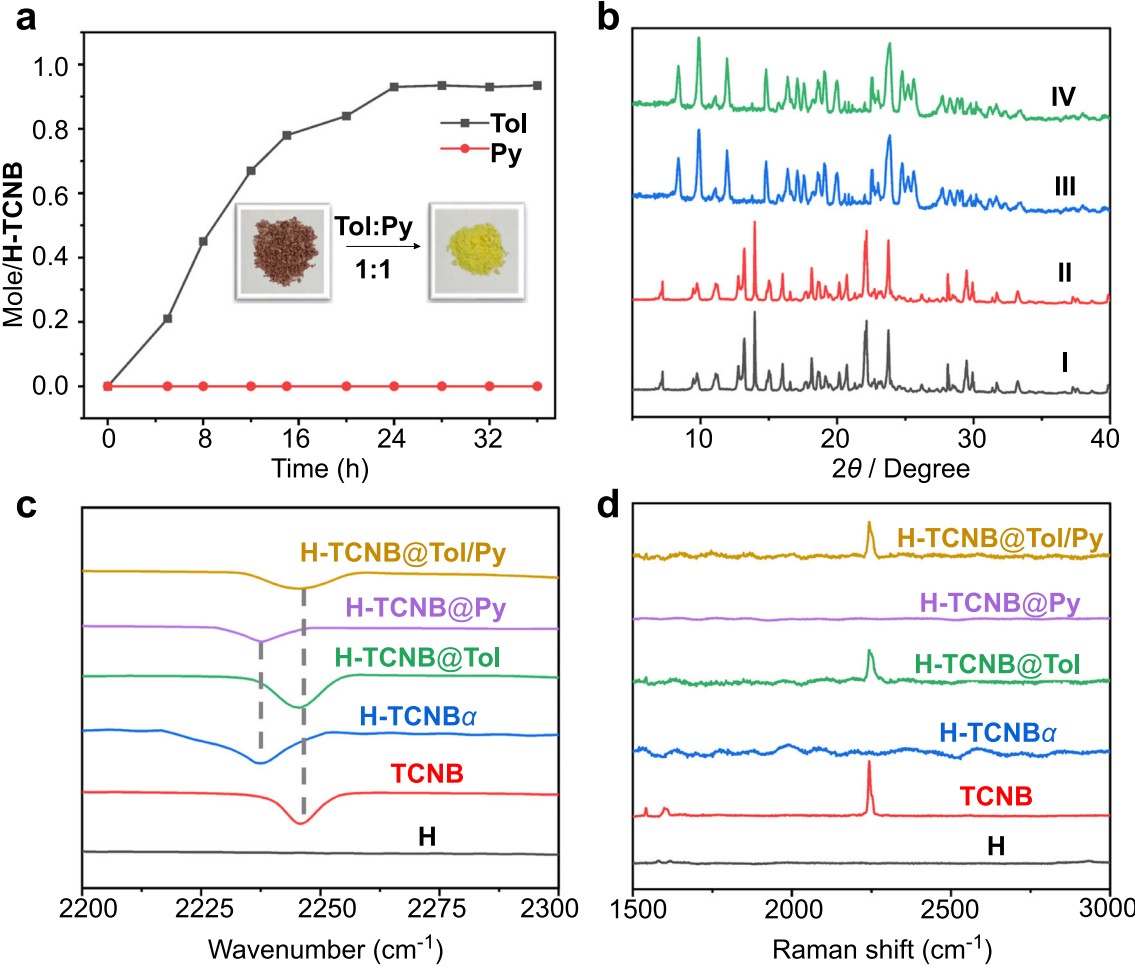

**Fig. 9 | Vapochromic behaviors H-TCNBα toward the vapors of Tol and Py.** **a** Time-dependent solid-vapor sorption plots of H-TCNBα for Tol and Py mixture vapors (*v:v* = 1:1). Inset showed the color change of H-TCNBα after adsorption of Tol and Py mixture vapors. **b** PXRD patterns of H-TCNBα: (I) original H-TCNBα; (II) after adsorption of Py vapor; (III) after adsorption of Tol vapor; (IV) after adsorption of Tol and Py mixture vapors. **c** Partial FT-IR spectra of H, TCNB, H-TCNBα, H-TCNB@Tol, H-TCNB@Py and H-TCNB@Tol/Py. **d** Partial Raman spectra of H, TCNB, H-TCNBα, H-TCNB@Tol, H-TCNB@Py and H-TCNB@Tol/Py.

could be visually monitored. Furthermore, H-TCNBα could be used as an effective adsorbent to improve the purity of Cy from 98.00% to 99.72% and the purity of Py from 97.94% to 99.35%. Single crystal X-ray diffraction, PXRD analyses and theoretical calculations proved that the mechanism of adsorption and vapochromic behavior of H-TCNBα originated from the solid-state structural transformation when H-TCNBα captured different vapors. The reversible transformation between adsorption and desorption made H-TCNBα highly recyclable. Considering the high separation efficiency, excellent vapochromic behavior and outstanding recycling performance, hybrid[4]arene-based co-crystal materials hold tremendous potential for advancement in the fields of petrochemical industry, environmental monitoring and gas sensing.

## Methods

### Materials
Starting materials and reagents including TCNB (98%), Bz (99%), Cy (99%), Tol (99%) and Py (99%) were purchased from Energy Chemical supplier and used without further purification unless stated otherwise.

### Characterization
Solution $^1$H NMR spectra were recorded at 500 or 600 MHz using a Bruker Avance 500 or 600 NMR spectrometer. Powder X-ray diffraction (PXRD) data were collected on a Rigaku Ultimate-IV X-ray diffractometer operating at 40 kV/30 mA using the Mo Kα line

($\lambda = 1.5418$ Å). Data were measured over the range of 5 – 40° in 5°/min steps over 7 min. Thermogravimetric analysis (TGA) was carried out using a Q5000IR analyzer (TA Instruments) with an automated vertical overhead thermobalance. The samples were heated at 10 °C/min using $N_2$ as the protective gas. Single crystal X-ray diffraction data were collected on a Bruker D8 VENTURE CMOS X-ray diffractometer with graphite monochromated Mo Kα radiation ($\lambda = 0.71073$ Å). Diffuse reflectance spectra were recorded with a SHIMADZU UV-2550 spectrometer. UV-*vis* absorption spectra were recorded using a Perki-nElmer Lambda 35 UV-*vis* spectrophotometer. Differential Scanning Calorimetric study (DSC) was carried out using a DSC Q100 analyzer (TA Instruments). The samples were heated at 10 °C/min using $N_2$ as the protective gas. The FT-IR spectra were measured on a Perkin Elmer 480 FT-IR spectrophotometer (KBr pellet). The Raman spectra were measured on a Horiba scientific-LabRAM HR evolution.

### Single crystal growth
5 mg of dry hybrid[4]arene (H) and 1 mg of dry TCNB powder were dissolved in a small vial where 0.5 mL of $CH_2Cl_2$ was added and the small vial was put in a 20 mL vial containing 2 mL of *n*-hexane. Single crystals of $CH_2Cl_2$-loaded H-TCNB were grown by evaporating *n*-hexane into a $CH_2Cl_2$ solution of H with equimolar TCNB. Orange crystals were got after 2 days. Single crystals of H-TCNB@Bz were grown by slow evaporation: 5 mg of dry H and 1 mg of dry TCNB powder were put in a small vial where 1 mL of Bz was added. The resultant

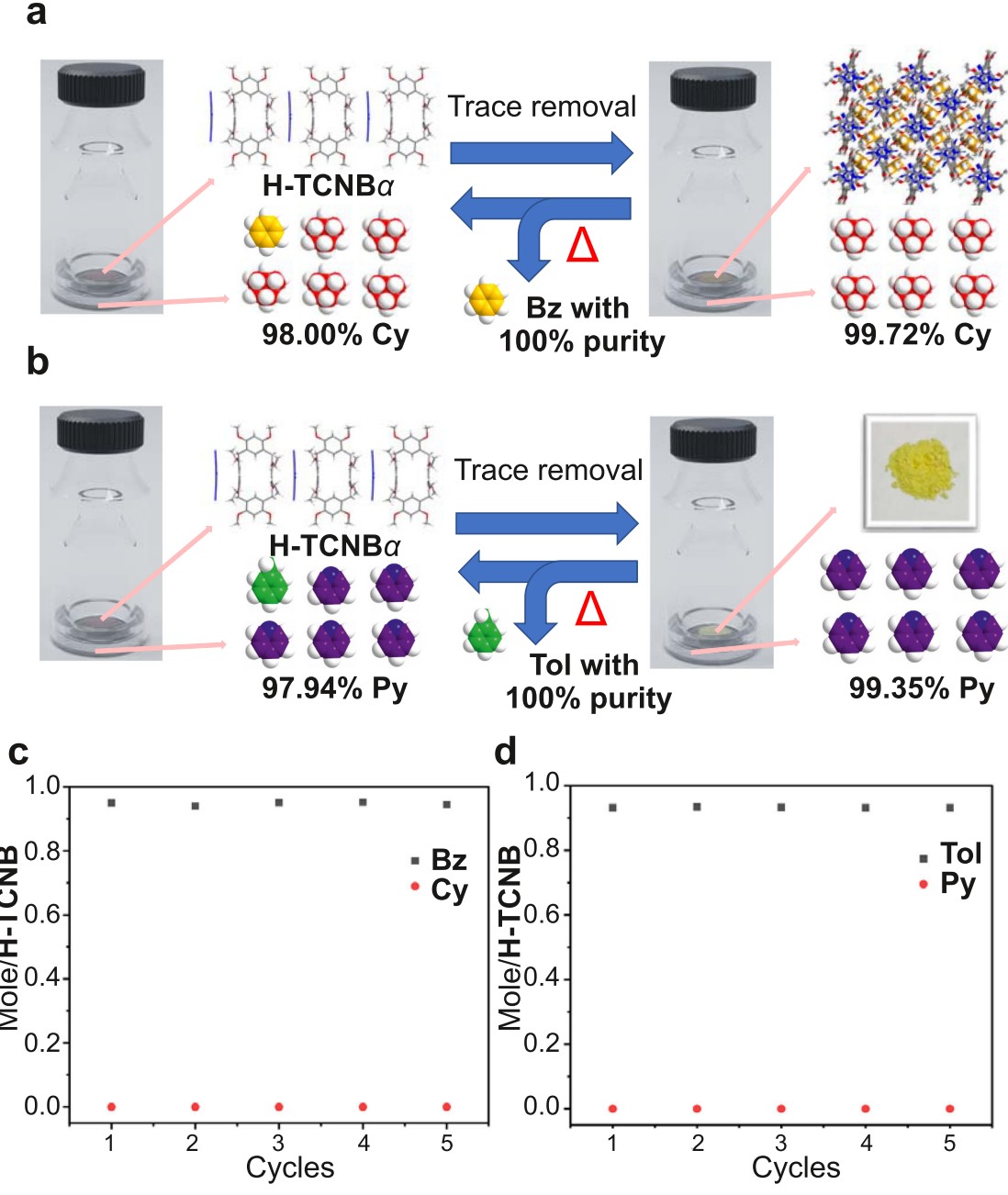

**Fig. 10 | Purification and recyclability experiments.** Schematic representation of purification of (**a**) Cy and (**b**) Py by H-TCNBα. Relative uptakes of **c** Bz and Cy vapors, **d** Tol and Py vapors by H-TCNBα after 5 cycles.

transparent solution was allowed to evaporate slowly to give nice colorless crystals in 2 to 3 days.

## Theoretical calculations

All calculations were performed by density functional theory (DFT) using the B3LYP hybrid function combined with 6-31 G (d,p) basis set under Gaussian G09. Using single crystal as input files, independent gradient model (IGM) analyses were carried out by Multiwfn 3.6 program through function 20 (visual study of weak interaction) and visualized by Visual Molecular Dynamics software.

## Gas chromatographic analysis

Gas Chromatography (GC) Analysis: GC measurements were carried out using an Agilent 7890B instrument configured with an FID detector and a DB-624 column (30 m × 0.53 mm × 3.0 $\mu$m). Samples were analyzed

using headspace injections and were performed by incubating the sample at 100 °C for 10 min followed by sampling 1 mL of the head-space. The following GC method was used: the oven was programmed from 50 °C, and ramped in 10 °C min⁻¹ increments to 150 °C with 15 min hold; the total run time was 25 min; the injection temperature was 250 °C; the detector temperature was 280 °C with nitrogen, air, and make-up flow rates of 35, 350, and 35 mL min⁻¹, respectively.

## Supplementary information

The online version contains supplementary material available at https://doi.org/10.1038/xxxxx.

## Data availability

All data supporting the findings of this study are available from the article and its Supplementary Information or available from the

corresponding author. The X-ray crystallographic coordinates for structures reported in this study have been deposited at the Cambridge Crystallographic Data Centre (CCDC), under deposition numbers 2244801, 2244721 and 2244722. These data can be obtained free of charge from The Cambridge Crystallographic Data Centre via www.ccdc.cam.ac.uk/data_request/cif.

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

## Acknowledgements

This work was supported by the National Natural Science Foundation of China (22101043), the Fundamental Research Funds for the Central Universities (N2205013, N232410019), the Opening Fund of State Key Laboratory of Heavy Oil Processing (SKLHOP202203006) and Northeastern University.

## Author contributions

J.Z. conceived and designed the project. J.C. conducted the synthesis and measurements, and wrote the manuscript. J.C. analyzed the data with the help from W.Z., W.Y., F.X., H.H., M.L., Q.D., J.H., M.W., G.Y. and J.Z. J.Z. reviewed and edited the manuscript. All authors discussed the results and commented on the manuscript.

## Competing interests

The authors declare no competing interests.
