## [Peer Review File · Nature Communications]

Separation of benzene and toluene associated with vapochromic behaviors by hybrid[4]arene-based co-crystalsREVIEWER COMMENTS

Reviewer #1 (Remarks to the Author):

In this work, Zhou et al. reported a new hybrid[4]arene-based charge transfer co-crystal, which exhibited vapochromic behaviors for benzene and toluene vapors. Moreover, a process of selective adsorption of benzene/cyclohexane and toluene/pyridine was visually observed, enabling the removal of trace amount of benzene and toluene from the corresponding mixture. The experimental section is also logical and well demonstrated. However, there have been many reports on the adsorptive separation of benzene/cyclohexane or toluene/pyridine by NACs, and high selectivity has also been achieved (DOI: 10.1039/d2cs00856d). Therefore, I don't think this work meets the novelty requirements and the high standard of Nat. Commun., neither in terms of the co-crystal design nor the application of adsorptive separation. Its publication in more special journals is suggested.

1. Some of references (such as ref. 13, 15, 17, 18, and 47) are actually inconsistent with the contents described in this work.
2. On line 179, the described content does not match the given figure.
3. On line 192, the "redshift" should be corrected to the "blue shift".
4. To gain a deeper understanding on the selective adsorption of toluene/pyridine, the corresponding crystal structure should be provided.
5. The tables in supplementary Figures 59-66 should be reported in English.

Reviewer #2 (Remarks to the Author):

In this manuscript, Zhou et al. described the highly selective uptake of benzene and toluene vapors from cyclohexane and pyridine using co-crystal of a novel hybridarene and tetracyanobenzene as a separating agent.

Furthermore, this co-crystal has the feature that the adsorption and desorption can be noticed by the color of the crystal due to the vaporchromic behavior among the hybridarene, tetracyanobenzene or/and the vapor molecules to be taken up by the charge transfer interaction. These phenomena have been demonstrated in a number of investigations and are evaluated to be chemically reliable.

In particular, the fact that benzene and toluene contained in trace amounts (about 2%) can be made highly pure to over 99% is a very intriguing result.

However, while the importance of the industrial need for high-purity separation is understandable, the reasons for selecting the vapor organic compounds to be separated are somewhat abstract, and more emphasis needs to be placed on the industrial urgency of why they must be separated. Also, the advantages of using new cyclic compounds remain unclear, and further explanation is needed.

As aforementioned above, I recommend that this manuscript be published in Nature Communications with the following modifications.

- (1) Separation of toluene and pyridine

Pyridine, which exhibits basicity, is easily soluble in water, and separation of toluene and pyridine using their dissolution in water is considered possible. In terms of industrial production, a pyridine removal process with water is expected to increase subsequent water treatment steps, but it appears to be a much more productive method than purification from a vapor process. Appropriate papers and literature regarding evidence of pyridine mixture in toluene in industrial production should be cited. In other words, the reader would not understand the importance of this study dealing with the purification of toluene/pyridine without more reference to the industrial challenges posed by pyridine mixing. The authors should revise the points advised above in order to appear to the readers and make them understand the importance of this study.

(2) Advantages of using a hybridarene

The authors have a newly synthesized hybridarene of macrocyclic compounds. On the other hand, the separation of benzene/cyclohexane or toluene/pyridine does not utilize the cavity of this characteristic macrocyclic compound itself. The concept of molecular recognition chemistry is to the study of extracting the characteristics of proper molecular recognition by utilizing the cavity in the macrocyclic compounds to enable selective molecular recognition. In other words, the results of this study suggest that it is possible to produce the properties shown in this study with acyclic compounds itself. Hence, I assume that the reader does not understand why the employment of hybridarene is important in this study. I think that the reasons (advantages) why the hybridarene should be more elaborated (e.g., the structure that promotes charge-transfer interactions can only be achieved with a macrocyclic structure).

(3) Revision of the paper

Line 220—221

In text: (Supplementary Fig. 36).

Is it correct the number of Supplementary Figure?

Line 221

...(Fig. 2b and Supplementary Fig. 45).

I think no need "Supplementary Fig. 45"

Line 276-277

In Fig. 6f, it is not clear which part of H is electron-rich. A precise diagram should be prepared.

Line 279

due to the size matching....

It is not clear what this statement means. What kind of space is formed and how large molecules fit in the crystal?

Line 286

green regions

Which diagram does this refer to?

I could not find Supplementary Figures in the main text (for example, Supplementary Figures 46, 48, 53, and so on). Additionally, there are also some inadequate locations for the figure headings inserted in the

text (for example, Fig. 6e, etc.). Because of this, I had some difficulties understanding what the authors intended to mean in the text.

Reviewer #3 (Remarks to the Author):

Comments to the Authors:

In this manuscript, an environmentally friendly and energy-saving adsorptive separation method has been developed, which realizes the perfect separation of benzene and toluene associated with vaporochromic behaviors by novel hybrid[4]arene-based co-crystals. The visual adsorptive separation based on co-crystal materials provides a convenient strategy for the separation of volatile organic compounds with different electronegativity. Overall, this work is novel and solid, and this well-written manuscript will be of great interest to the broad readership of Nature Communications. Thus, I recommend publication pending revisions stated below:

1. "Supplementary Fig. S29" should be changed to "Supplementary Fig. 29".
2. "Fig.2b and Supplementary Fig. 45" should be changed to "Fig. 2b and Supplementary Fig. 45".
3. "Supplementary Fig. S49" should be changed to "Supplementary Fig. 49".
4. "Supplementary Fig.67-72" should be changed to "Supplementary Fig. 67-72".
5. Ref. 12: Can this reference be updated? If so, please provide the relevant information such as year, volume and page or article numbers as appropriate.
6. Introduction, the authors mentioned "macrocycle-based vapochromic materials", the pionnering papers close related to this emerging field should be cited: Nat. Commun. 14, 5954 (2023); Angew. Chem. Int. Ed. 2023, 62, e202218142 ;Angew. Chem. Int. Ed. 61, e202210579 (2022).

RESPONSE TO REVIEWERS' COMMENTS

We very much appreciate all the valuable comments raised by these three reviewer because these comments are very helpful for us to improve our manuscript. We made the following corrections and improvements point by point in response to these comments.

Reply to Reviewer 1:

In this work, Zhou et al. reported a new hybrid[4]arene-based charge transfer co-crystal, which exhibited vapochromic behaviors for benzene and toluene vapors. Moreover, a process of selective adsorption of benzene/cyclohexane and toluene/pyridine was visually observed, enabling the removal of trace amount of benzene and toluene from the corresponding mixture. The experimental section is also logical and well demonstrated. However, there have been many reports on the adsorptive separation of benzene/cyclohexane or toluene/pyridine by NACs, and high selectivity has also been achieved (DOI: 10.1039/d2cs00856d). Therefore, I don't think this work meets the novelty requirements and the high standard of Nat. Commun., neither in terms of the co-crystal design nor the application of adsorptive separation. Its publication in more special journals is suggested.

Thank you for your sincere and precious opinions. We would like to clarify and further emphasize the impact and great potential of this work.

First, for the first time, we synthesized a novel macrocycle hybrid[4]arene, and demonstrated the perfect separation of benzene and toluene associated with vapochromic behaviors by hybrid[4]arene-based co-crystals. This is truly innovative and important. The development of new macrocyclic hosts with excellent structural and functional characteristics is an eternal theme in supramolecular chemistry. Although great progress has been made in the research of macrocyclic hosts composed of identical repeating units in recent years, it is still a great challenge to effectively design and synthesize new functional macrocyclic hosts with different types of repeating units.

Second, in recent years, co-crystals have provided a unique strategy for the preparation of multifunctional vapochromic materials. Hybrid[4]arene-based co-crystals show the efficient separation of benzene from benzene/cyclohexane mixtures as well as toluene from toluene/pyridine mixtures with purities reaching 100%, accompanied by color changes from brown to yellow. The process of adsorptive separation can be visually monitored. Furthermore, hybrid[4]arene-based co-crystals can be used as an effective adsorbent to improve the purity of cyclohexane from 98.00% to 99.72%, and the purity of pyridine from 97.94% to 99.35%. To our knowledge, this is the first case of perfect separation of benzene and toluene associated with vapochromic behaviors by hybridarenes-based co-crystals.

Third, the adsorption and separation of benzene/cyclohexane or toluene/pyridine by NACs are important, which hold great potential in practical applications. Hybrid[4]arene-based co-crystals, as a kind of adsorption material with both the features of selective

molecular recognition and vapochromism, deserve to be appreciated in industry. This discovery is intriguing and enlightening for the general community from macrocycle co-crystals, and will promote the development of materials with vapochromic behaviors in supramolecular science.

Overall, we strongly believe that this study has potential interest to the broad readership of *Nature Communications*, especially those working on crystal engineering and supramolecular materials, and could also inspire researchers to continue to explore possibilities of multifunctional sensing/adsorbing materials.

1. *Some of references (such as ref. 13, 15, 17, 18, and 47) are actually inconsistent with the contents described in this work.*

Thank you very much. The corresponding corrections were all made.

2. *On line 179, the described content does not match the given figure.*

Thank you very much. This was corrected. We changed this sentence to: “In addition, the uptake of **Bz** vapor was calculated as one **Bz** vapor per **H-TCNB** molecule at saturation by ¹H NMR spectrum (Fig. 4a, black line; Supplementary Fig. 31). The uptake of **Tol** vapor was calculated as 0.9 **Tol** vapor per **H-TCNB** molecule at saturation by ¹H NMR spectrum (Fig. 4d, black line; Supplementary Fig. 33)”.

3. *On line 192, the “redshift” should be corrected to the “blue shift”.*

Thank you very much. This was done.

4. *To gain a deeper understanding on the selective adsorption of toluene/pyridine, the corresponding crystal structure should be provided.*

Thank you very much for your kind advice. Various attempts have been made to obtain single crystals of **H-TCNB@Tol**, but failed. In order to gain a deeper understanding on the selective adsorption of toluene/pyridine, DFT calculations were performed at the wB97XD/6–311G(d) theoretical level, revealing the binding modes of **H-TCNB** to toluene (**Tol**) and pyridine (**Py**). In the calculated binding mode of **H-TCNB@Py**, the centroid-centroid distance of **H** with **TCNB** was 3.297 Å, which was similar to that of **H-TCNB α** (Supplementary Fig. 59). In the calculated binding mode of **H-TCNB@Tol**, the centroid-centroid distance of **H** with **TCNB** was 3.596 Å, which was different from that of **H-TCNB α** . The results showed that the structure of **H-TCNB α** changed after exposure to **Tol** vapor. The calculated energy values showed that the formation of **H-TCNB@Tol** was thermodynamically feasible ($\Delta H_{\text{Sol}}^{298} = -43.33$ kJ/mol, $\Delta G_{\text{Sol}}^{298} = -38.33$ kJ/mol) (Supplementary Table 1). Compared with **H-TCNB@Tol**, the thermodynamic energy values of **H-TCNB@Py** were higher ($\Delta H_{\text{Sol}}^{298} = 20.16$ kJ/mol, $\Delta G_{\text{Sol}}^{298} = 12.20$ kJ/mol), illustrating that the thermodynamic structure of **H-TCNB@Tol** was more stable. In agreement with the experimentally observed selectivity, the thermodynamic calculation results showed that the

selectivity arose from the thermodynamic stability of the new formed co-crystals structure of **H-TCNB α** upon capturing the **Tol**.

Supplementary Figure 59. The centroid-centroid distance of **H** with **TCNB** in **a H-TCNB α** (3.267 Å) and calculated binding modes of **b H-TCNB@Tol** (3.596 Å) and **c H-TCNB@Py** (3.587 Å).

Supplementary Table 1. Thermodynamic parameters of calculated binding modes of **H-TCNB@Tol** and **H-TCNB@Py** at 298 K.

	ΔH (kJ/mol)	ΔG (kJ/mol)
H-TCNB@Tol	-43.33	-38.33
H-TCNB@Py	20.16	12.20

5. The tables in supplementary Figures 59-66 should be reported in English.

Thank you very much. These were all done.

Reply to Reviewer 2:

1. *Separation of toluene and pyridine*

Pyridine, which exhibits basicity, is easily soluble in water, and separation of toluene and pyridine using their dissolution in water is considered possible. In terms of industrial production, a pyridine removal process with water is expected to increase subsequent water treatment steps, but it appears to be a much more productive method than purification from a vapor process. Appropriate papers and literature regarding evidence of pyridine mixture in toluene in industrial production should be cited. In other words, the reader would not understand the importance of this study dealing with the purification of toluene/pyridine without more reference to the industrial challenges posed by pyridine mixing.

Thank you very much for your kind advices. The importance of this study dealing with the purification of toluene/pyridine was added as follows: In industry, toluene (**Tol**) is mainly obtained from coal and petroleum through catalytic reforming, hydrocarbon pyrolysis and coking (*Ind. Eng. Chem. Res.* **56**, 11894–11902 (2017); *Energy* **50**, 103–109 (2013)). However, the **Tol** obtained from the coal coking process often contains a small quantity of pyridine (**Py**) (*Ind. Eng. Chem. Res.* **54**, 7715–7727

(2015)). In order to obtain high-purity **Tol**, the small amount of **Py** present in the **Tol** during coal coking process must be removed (*Angew. Chem. Int. Ed.* **61**, e202207209 (2022)). These publications were cited as refs. 26, 27, 28 and 29.

2. Advantages of using a hybridarene

The authors have a newly synthesized hybridarene of macrocyclic compounds. On the other hand, the separation of benzene/cyclohexane or toluene/pyridine does not utilize the cavity of this characteristic macrocyclic compound itself. The concept of molecular recognition chemistry is to the study of extracting the characteristics of proper molecular recognition by utilizing the cavity in the macrocyclic compounds to enable selective molecular recognition. In other words, the results of this study suggest that it is possible to produce the properties shown in this study with acyclic compounds itself. Hence, I assume that the reader does not understand why the employment of hybridarene is important in this study. I think that the reasons (advantages) why the hybridarene should be more elaborated (e.g., the structure that promotes charge-transfer interactions can only be achieved with a macrocyclic structure).

Thank you very much for your kind comments. The development of new macrocyclic hosts with structural and functional characteristics is an eternal theme in supramolecular chemistry. Although great progress has been made in the research of macrocyclic hosts composed of the same repeating units in recent years, it is still a great challenge to effectively design and synthesize new functional macrocyclic hosts with different types of repeating units. Hybridarenes are synthesized by two or more different types of units, which broadens the scope for synthesizing macrocyclic hosts with varying shapes, cavity sizes and flexibilities. In this manuscript, for the first time, we report the synthesis and structure of hybrid[4]arene, which is made of two 1,3-dimethoxybenzene units and two 2,6-diethoxynaphthalene units linked by methylene bridges. Hybrid[4]arene exhibits a neat quadrilateral-prismatic structure. In particular, the naphthalene units impart electronegativity to hybrid[4]arene, enabling hybrid[4]arene to form stable co-crystals with electron-deficient guests.

On the other hand, in recent years, macrocycle co-crystals have provided a unique strategy for the preparation of multifunctional vapochromic materials. Hybrid[4]arene-based co-crystals display efficient separation of benzene from benzene/cyclohexane mixtures as well as toluene from toluene/pyridine mixtures with purities reaching 100%, accompanied by color changes from brown to yellow. This discovery is intriguing and enlightening for the general community from macrocycle co-crystals, and will promote the development of materials with vapochromic behaviors in supramolecular science.

Moreover, in order to verify that the vapochromic behaviors of co-crystals in this study can only be achieved with a macrocyclic structure, we added the corresponding control experiments as follows:

The 2,6-diethoxynaphthalene (**DON**) and the 1,2,4,5-tetracyanobenzene (**TCNB**) formed co-crystals (denoted as **DON-TCNB**) in CH₂Cl₂ through charge-transfer interaction (Supplementary Figs. 74 and 75). The complexation of **DON** with **TCNB**

was investigated by NMR and UV-*vis* absorption spectroscopy. After adding **TCNB** to **H** in CD_2Cl_2 , the signals related to the proton H_a on **TCNB** and the proton H_1 on **DON** shifted upfield ($\Delta\delta = -0.05$ ppm and -0.02 ppm, respectively) (Supplementary Fig. 76). After adding **TCNB** to **DON** in CH_2Cl_2 , the color of the solution of **DON** changed from colorless to purple, and a new absorption band appeared at 375 nm in the UV-*vis* absorption spectrum. These results confirmed the existence of charge-transfer interaction between **DON** and **TCNB** (Supplementary Fig. 77).

Supplementary Figure 74. a Chemical structures of **DON** and **TCNB**. **b** Images of **DON**, **TCNB**, and **DON-TCNB**.

Supplementary Figure 76. Partial ^1H NMR spectra (600 MHz, CD_2Cl_2 , 293 K) of **a** free TCNB (10.0 mM), **b** DON (10.0 mM) and TCNB (10.0 mM), and **c** free DON (10.0 mM).

Supplementary Figure 77. **a** Images: (I) DON (5.00 mM) in CH_2Cl_2 ; (II) DON (5.00 mM) and TCNB (5.00 mM) in CH_2Cl_2 ; (III) TCNB (5.00 mM) in CH_2Cl_2 . **b** UV-*vis* spectra: DON (5.00 mM); DON (5.00 mM) and TCNB (5.00 mM); TCNB (5.00 mM).

To obtain adsorptive and vaporchromic materials, **DON-TCNB** was activated by heating at 80 °C for 10 h under vacuum (named **DON-TCNB α**). ^1H NMR and TGA verified that the molar ratio of **DON** to **TCNB** was 1:1 (Supplementary Figs. 78-80). The PXRD pattern of **DON-TCNB α** was similar to that of **DON-TCNB** (Supplementary Fig. 81), indicating that the structure of **DON-TCNB** was maintained

after activation. The Fourier transform infrared spectrum (FT-IR) of **TCNB** showed a typical stretching vibration peak associated with the CN group at 2247 cm^{-1} (Supplementary Fig. 82). However, the CN peak in **DON-TCNB α** was redshifted. In addition, the diffuse reflectance spectrum of **DON-TCNB α** exhibited an obvious broad band at 500 nm (Supplementary Fig. 83), while no absorption was observed in neither **DON** nor **TCNB**, demonstrating the charge-transfer interaction between **TCNB** and **DON**.

Supplementary Figure 78. Images of **DON-TCNB** and **DON-TCNB α** .

Supplementary Figure 79. ^1H NMR spectrum (600 MHz, CD_2Cl_2 , 293 K) of **DON-TCNB α** .

Supplementary Figure 80. Thermogravimetric analysis of **DON-TCNB α** .

Supplementary Figure 81. PXRD patterns and images of (I) **DON-TCNB** and (II) **DON-TCNB α** .

Supplementary Figure 82. FT-IR spectra of DON, TCNB and DON-TCNB α .

Supplementary Figure 83. Diffuse reflectance spectra and images of DON-TCNB α , DON and TCNB.

In addition, the adsorptive and vapochromic behaviors of single-component **Bz**, **Cy**, **Tol** and **Py** vapors by **DON-TCNB α** were studied. The color of **DON-TCNB α** did not change when exposed to **Bz**, **Cy**, **Tol** or **Py** vapor for 24 h, respectively. The uptake of **Bz**, **Cy**, **Tol** or **Py** vapor by **DON-TCNB α** was negligible by ^1H NMR spectra (Supplementary Figs. 84-88). Besides, the PXRD patterns of **DON-TCNB α** did not change after adsorption of **Bz**, **Cy**, **Tol** or **Py** vapor (Supplementary Fig. 89). These results indicated that **DON-TCNB α** could not capture **Bz**, **Cy**, **Tol** or **Py** vapor.

Supplementary Figure 84. ^1H NMR spectrum (600 MHz, CD_2Cl_2 , 293 K) of **DON-TCNB α** after adsorption of **Bz** vapor for 24 h.

Supplementary Figure 85. ¹H NMR spectrum (600 MHz, CD₂Cl₂, 293 K) of **DON-TCNB α** after adsorption of **Cy** vapor for 24 h.

Supplementary Figure 86. ¹H NMR spectrum (600 MHz, CD₂Cl₂, 293 K) of **DON-TCNB α** after adsorption of **Tol** vapor for 24 h.

Supplementary Figure 87. ¹H NMR spectrum (600 MHz, CD₂Cl₂, 293 K) of DON-TCNB α after adsorption of **Py** vapor for 24 h.

Supplementary Figure 88. ¹H NMR spectra (600 MHz, CD₂Cl₂, 293 K) of DON-TCNB α : **a** original DON-TCNB α ; after adsorption of **b Bz** vapor; **c Cy** vapor; **d Tol** vapor and **e Py** vapor for 24 h, respectively.

Supplementary Figure 89. PXRD patterns of **DON-TCNB α** : (I) original **DON-TCNB α** ; (II) after adsorption of **Bz** vapor; (III) after adsorption of **Cy** vapor; (IV) after adsorption of **Tol** vapor; (V) after adsorption of **Py** vapor.

The 1,3-dimethoxybenzene (**DOB**) and the 1,2,4,5-tetracyanobenzene (**TCNB**) formed co-crystals (denoted as **DOB-TCNB**) in CH_2Cl_2 through charge-transfer interaction. **DOB-TCNB** was investigated by NMR and UV-*vis* absorption spectroscopy. After adding **TCNB** to **H** in CD_2Cl_2 , the signal related to the proton H_a on **TCNB** shifted upfield ($\Delta\delta = -0.03$ ppm) (Supplementary Figs. 90 and 91). After adding **TCNB** to **DOB** in CH_2Cl_2 , the color of the solution of **DOB** changed from colorless to green, and a new absorption band appeared at 350 nm in the UV-*vis* absorption spectrum. These results confirmed the existence of charge-transfer interaction between **DOB** and **TCNB** (Supplementary Fig. 92).

Supplementary Figure 90. ¹H NMR spectrum (600 MHz, CD₂Cl₂, 293 K) of **DOB-TCNB**.

Supplementary Figure 91. Partial ¹H NMR spectra (600 MHz, CD₂Cl₂, 293 K) of **a** free **TCNB** (10.0 mM), **b** **DOB** (10.0 mM) and **TCNB** (10.0 mM), and **c** free **DOB** (10.0 mM).

Supplementary Figure 92. **a** Images: (I) **DOB** (5.00 mM) in CH₂Cl₂; (II) **DOB** (5.00 mM) and **TCNB** (5.00 mM) in CH₂Cl₂; (III) **TCNB** (5.00 mM) in CH₂Cl₂. **b** UV-*vis* spectra: **DOB** (5.00 mM); **DOB** (5.00 mM) and **TCNB** (5.00 mM); **TCNB** (5.00 mM).

Various attempts were made to obtain the co-crystals of **DOB-TCNB**, and green crystals were obtained. When washed with ethanol, the color of the crystals changed from green to colorless (Supplementary Fig. 93). The colorless crystals were activated by heating at 80 °C for 10 h under vacuum (named **DOB-TCNB α**). ¹H NMR, PXRD and FT-IR verified that the colorless crystals were **TCNB** (Supplementary Figs. 94-96).

Above all, the adsorptive and vaporchromic behaviors of hybrid[4]arene-based co-crystals can only be achieved with a macrocyclic structure.

Supplementary Figure 93. Images of **DOB-TCNB** and **DOB-TCNB α** .

Supplementary Figure 94. ¹H NMR spectrum (600 MHz, CD₂Cl₂, 293 K) of DOB-TCNB α .

Supplementary Figure 95. PXRD patterns: (I) TCNB; (II) DOB-TCNB α .

Supplementary Figure 96. FT-IR spectra of **DOB**, **TCNB** and **DOB-TCNB α** .

3. *Revision of the paper*

Line 220–221

In text: (Supplementary Fig. 36).

Is it correct the number of Supplementary Figure?

Thank you very much. This was corrected.

4. *Line 221*

...(Fig. 2b and Supplementary Fig. 45).

I think no need “Supplementary Fig. 45”

Thank you. This was done.

5. *Line 276-277*

*In Fig. 6f, it is not clear which part of **H** is electron-rich. A precise diagram should be prepared.*

Thank you very much. As shown in Fig. 6, the electron-rich area on **H** referred to the methoxyl group. We changed “the electron-rich area on **H**” to “the electron-rich area of methoxyl group on **H**”.

Fig. 6. ESP maps of **H-TCNB α** and **H-TCNB@Bz**. The area of red circles represents the change in electron density on **H-TCNB α** after the capture of **Bz**.

6. *Line 279*

due to the size matching....

It is not clear what this statement means. What kind of space is formed and how large molecules fit in the crystal?

Thank you very much for your kind comments. The aromatic area of **Bz** displayed electronegativity, while the outer edge of **Bz** showed electropositivity (Scheme 1). In addition, the methoxy group on **H** of **H-TCNB α** was electronegative, and the hydrogen atom on **TCNB** of **H-TCNB α** was electron-deficient. Therefore, **H-TCNB α** was prone to form charge-transfer complex with **Bz**. We changed “due to the size matching” to “due to charge-transfer interaction”.

7. *Line 286*

green regions

Which diagram does this refer to?

Thank you very much. The “green region” was shown in Fig. 7. Green regions referred to multiple non-covalent interactions between **H** and **TCNB**.

8. *I could not find Supplementary Figures in the main text (for example, Supplementary Figures 46, 48, 53, and so on). Additionally, there are also some inadequate locations for the figure headings inserted in the text (for example, Fig. 6e, etc.). Because of this, I had some difficulties understanding what the authors intended to mean in the text.*

Thank you very much. These were all done. We checked the full text carefully and added relevant Supplementary Figures and adjusted the position of figure headings in the main text.

Reply to Reviewer 3:

1. “Supplementary Fig. S29” should be changed to “Supplementary Fig. 29”.

Thank you very much. This was done.

2. *“Fig.2b and Supplementary Fig. 45” should be changed to “Fig. 2b and Supplementary Fig. 45”.*

Thank you very much. This was done.

3. *“Supplementary Fig. S49” should be changed to “Supplementary Fig. 49”.*

Thank you very much. This was done.

4. *“Supplementary Fig.67-72” should be changed to “Supplementary Fig. 67-72”.*

Thank you very much. This was done.

5. *Ref. 12: Can this reference be updated? If so, please provide the relevant information such as year, volume and page or article numbers as appropriate.*

Thank you very much. This reference has been updated.

Ref. 12: *Chem. Soc. Rev.* **52**, 6075–6119 (2023).

6. *Introduction, the authors mentioned “macrocycle-based vapochromic materials”, the pionnering papers close related to this emerging field should be cited: Nat. Commun. 14, 5954 (2023); Angew. Chem. Int. Ed. 2023, 62, e202218142; Angew. Chem. Int. Ed. 61, e202210579 (2022).*

Thank you very much. These publications were cited as refs. 16, 17 and 18.

REVIEWERS' COMMENTS

Reviewer #1 (Remarks to the Author):

The authors have revised the manuscript carefully according to the comments and suggestions of the reviewers, so publication is now suggested.

Reviewer #2 (Remarks to the Author):

I have again reviewed the revised paper and the author's comments. It has been carefully revised in response to all my comments. Based on the above, I recommend the publication of this paper in your esteemed journal.

Reviewer #3 (Remarks to the Author):

The authors have submitted a revised manuscript based on reviewers' comments, and I have carefully read it and the responding letter. The authors adequately responded to the reviewers' comments, and this paper is now acceptable.

RESPONSE TO REVIEWERS' COMMENTS

We very much appreciate all the valuable comments raised by these three reviewer because these comments are very helpful for us to improve our manuscript. We made the following corrections and improvements point by point in response to these comments.

Reply to Reviewer 1:

The authors have revised the manuscript carefully according to the comments and suggestions of the reviewers, so publication is now suggested.

We greatly appreciate the reviewer's efforts on helping us improve the quality of our manuscript.

Reply to Reviewer 2:

I have again reviewed the revised paper and the author's comments. It has been carefully revised in response to all my comments. Based on the above, I recommend the publication of this paper in your esteemed journal.

We greatly appreciate the reviewer's efforts on helping us improve the quality of our manuscript.

Reply to Reviewer 3:

The authors have submitted a revised manuscript based on reviewers' comments, and I have carefully read it and the responding letter. The authors adequately responded to the reviewers' comments, and this paper is now acceptable.

We greatly appreciate the reviewer's efforts on helping us improve the quality of our manuscript.